# RSC-SNN: Exploring the Trade-off Between Adversarial Robustness and Accuracy in Spiking Neural Networks via Randomized Smoothing Coding

Keming Wu*
Chongqing University
Chongqing, China
wukemingcqu@gmail.com

Man Yao*
Institute of Automation, Chinese
Academy of Sciences
Beijing, China
man.yao@ia.ac.cn

Yuhong Chou
Xi'an Jiaotong University
Xi'an, China
yuhong_chou@outlook.com

Xuerui Qiu
Institute of Automation, Chinese
Academy of Sciences
Beijing, China
qiuxuerui2024@ia.ac.cn

Rui Yang
Software Security Technology
Company Ltd
Beijing, China
Yangrui@softsafe-tech.com

Bo Xu
Institute of Automation, Chinese
Academy of Sciences
Beijing, China
xubo@ia.ac.cn

Guoqi Li†
Institute of Automation, Chinese
Academy of Sciences
Beijing, China
guoqi.li@ia.ac.cn

## Abstract

Spiking Neural Networks (SNNs) have received widespread attention due to their unique neuronal dynamics and low-power nature. Previous research empirically shows that SNNs with Poisson coding are more robust than Artificial Neural Networks (ANNs) on small-scale datasets. However, it is still unclear in theory how the adversarial robustness of SNNs is derived, and whether SNNs can still maintain its adversarial robustness advantage on large-scale dataset tasks. This work theoretically demonstrates that SNN's inherent adversarial robustness stems from its Poisson coding. We reveal the conceptual equivalence of Poisson coding and randomized smoothing in defense strategies, and analyze in depth the trade-off between accuracy and adversarial robustness in SNNs via the proposed Randomized Smoothing Coding (RSC) method. Experiments demonstrate that the proposed RSC-SNNs show remarkable adversarial robustness, surpassing ANNs and achieving state-of-the-art robustness results on large-scale dataset ImageNet. Our open-source implementation code is available at *https://github.com/KemingWu/RSC-SNN*.

---

*Equal contribution.
†Corresponding author.

---

*MM '24, October 28-November 1, 2024, Melbourne, VIC, Australia*
© 2024 Copyright held by the owner/author(s). Publication rights licensed to ACM.
ACM ISBN 979-8-4007-0686-8/24/10
https://doi.org/10.1145/3664647.3680639

## CCS Concepts

• **Computing methodologies** → **Computer vision**; **Bio-inspired approaches**.

## Keywords

Spiking Neural Networks, Adversarial Learning, Randomized Smoothing

**ACM Reference Format:**
Keming Wu, Man Yao, Yuhong Chou, Xuerui Qiu, Rui Yang, Bo Xu, and Guoqi Li. 2024. RSC-SNN: Exploring the Trade-off Between Adversarial Robustness and Accuracy in Spiking Neural Networks via Randomized Smoothing Coding. In *Proceedings of the 32nd ACM International Conference on Multimedia (MM '24), October 28-November 1, 2024, Melbourne, VIC, Australia.* ACM, New York, NY, USA, 9 pages. https://doi.org/10.1145/3664647.3680639

## 1 Introduction

Owing to the distinctive event-driven nature [3] and remarkable biological plausibility [11], SNNs have gained recognition as the third generation of artificial neural networks [18, 25]. Compared with ANNs, SNNs employ discrete binary signals for information transfer among spiking neurons, where spikes are generated solely when the membrane potential surpasses the firing threshold. After deployment to neuromorphic chips [6, 20, 22, 37], SNNs have demonstrated their effectiveness and efficacy in a variety of scenarios, including static visual tasks[35, 36], dynamic visual processing [10, 38], speech classification [24, 39].

Direct [33] and Poisson coding [31] are two popular coding strategies for SNNs, which define how information is represented via spike patterns [7]. For SNNs, a static input $x \in \mathbb{R}^d$ needs to be converted into a time sequence input using coding strategies. Direct coding will repeatedly input $x$ for $T$ times. Poisson coding

uses frequency approximation to generate $T$ binary spikes $\{p_i\}_{i=1}^{T}$ so that the average number of spikes approximates the intensity of the pixel $\frac{1}{T}\sum_{i=1}^{T}p_i \approx x$. Coding methods play a crucial role in determining the network's computational efficiency and resilience to perturbations. Existing empirical studies have shown that the adversarial robustness of SNNs using Poisson coding is higher than that of ANNs, and the robustness decreases as the time step increases [28]. In contrast, SNNs using direct coding have poorer adversarial robustness than ANNs [15].

The impact of direct coding and Poisson coding on the adversarial robustness has not been systematically analyzed, which undermines the potential advantages of SNNs over ANNs in terms of adversarial robustness. Moreover, it is still unknown whether SNN can still maintain the adversarial robustness advantage on large-scale tasks, because previous work has only been verified on small datasets such as CIFAR-10/100 [4, 17, 21, 27]. We are interested in why the adversarial robustness of SNNs employing Poisson coding is stronger. We note that randomized smoothing and Poisson coding have similar features in enhancing adversarial robustness, although they may seem like two different approaches. Randomized smoothing builds a base classifier by introducing noise [5], in contrast, Poisson coding converts the input into a binary probability.

Inspired by this, we theoretically establish the connection between randomized smoothing and Poisson coding via analyzing the statistical characteristics of them. We found that Poisson coding shares fundamental statistical properties with randomized smoothing, such as expectation and variance, introducing similar noise smoothing. Since existing research has shown that randomized smoothing can bring certified adversarial robustness, based on this observation, we can understand why SNNs using Poisson coding have adversarial robustness. However, our theoretical analysis shows that SNNs with Poisson coding are greatly affected by perturbation while bringing about the problem of reduced clean accuracy. Therefore, it is urgent to establish a guiding principle for the trade-off between accuracy and robustness in the design of defense methods against adversarial examples in SNNs. We analyze in depth the trade-off between accuracy and adversarial robustness in SNNs via a novel Randomized Smoothing Coding (RSC) method, which significantly improves the adversarial robustness of SNNs. To further exploit the potential of this approach, we propose a new training method designed for RSC-SNN. Experimental results on extensive datasets show that randomized smoothing coding greatly enhances the adversarial robustness of SNNs. Simultaneously, The final results indicate a trade-off between accuracy and adversarial robustness, which is consistent with the conclusions of ANNs [29, 30, 41]. Furthermore, We also propose an empirical estimation method to quantify the trade-off called Quantification Trade-off Estimation (QTE) to help design defense methods with better trade-offs. The main contributions of our work are summarized as follows:

- We establish the connection between Poisson coding and randomized smoothing for the first time, which is a novel insight contributing to the field. Furthermore, we prove the conceptual equivalence of randomized smoothing and Poisson coding, which provides a theoretical foundation for the robustness of Poisson-encoded SNNs. We proposed a novel coding method called **randomized smoothing coding**.

- Observing the inherent clean accuracy drop caused by randomized smoothing coding, we propose a new training method called Efficient Randomized Smoothing Coding Training (**E-RSCT**) specifically for randomized smoothing coding.

- Experimental results show that RSC-SNNs show remarkable adversarial robustness in image recognition and achieves state-of-the-art results on datasets including large datasets Tiny-ImageNet, ImageNet, while achieving a better trade-off under the metric of Quantification Trade-off Estimation.

## 2 Background and Related Work

### 2.1 Spiking Neural Network

Spiking neurons are the basic units of SNNs, which are abstracted from the dynamics of biological neurons. The leaky-integrate-and-fire (LIF) neuron model is widely acknowledged as the simplest model among all popular neuron models while maintaining biological interpretability, in contrast to the many-variable and complex H-H model [14]. It also has a significantly lower computational demand [23, 26]. We adopt the LIF neuron model and translate it to an iterative expression with the Euler method [32]. Mathematically, the LIF-SNN layer can be described as an iterable version for better computational traceability:

$$\begin{cases} u_i^{(l)}[t+1] = h_i^{(l)}[t] + f(w^{(l)}, x_i^{(l-1)}[t]) \\ s_i^{(l)}[t] = \Theta(u_i^{(l)}[t+1] - \vartheta) \\ h_i^{(l)}[t+1] = \tau u_i^{(l)}[t+1](1 - s_i^{(l)}[t]), \end{cases} \quad (1)$$

where $\tau$ is the time constant, $t$ and $i$ respectively represent the indices of the time step and the $l$-th layer, $w$ denotes synaptic weight matrix between two adjacent layers, $f(\cdot)$ is the function operation stands for convolution (Conv) or fully connected (FC), $x$ is the input, and $\Theta(\cdot)$ denotes the Heaviside step function. When the membrane potential $u$ exceeds the firing threshold $\vartheta$, the LIF neuron will trigger a spike $S$. Moreover, $h$ represents the membrane potential after the trigger event which equals $\tau u$.

### 2.2 Adversarial Attacks

Adversarial attacks are designed to fool a model into incorrect predictions or outputs through carefully crafted inputs [12]. Given a classifier $f : \mathbb{R}^d \rightarrow \mathcal{Y}$, where $\mathcal{Y}$ is the set of class labels, the purpose of an adversarial perturbation $\delta$ is to make $f(x+\delta) \neq f(x)$, which can be formulated as an optimization problem:

$$\max_{\|\delta\|_p \leq \epsilon} \mathcal{L}(f(x+\delta), y), \quad (2)$$

where $f$ is the network under attack, $\mathcal{L}$ is the loss function, $x, y$ are the input and target output of the given network, respectively. $\epsilon$ is a parameter that limits the intensity of the perturbation so that it is not easily observed by the human eye. $\delta$ is the parameter we want to optimize. In this paper we mainly use two widely adopted gradient-based adversarial attacks: Fast Gradient Sign Method (FGSM) and Projected Gradient Descent method (PGD).

**FGSM.** As a simple but effective attack method [12], adversarial examples are generated based on the symbolic information of the gradient to maximize the loss of the perturbed $x + \delta$, which can be formulated as

$$x_{adv} = x + \epsilon \times \text{sign}\left(\nabla_x L(f(x, y))\right), \quad (3)$$

where $\epsilon$ denotes the strength of the attack.

**PGD.** As an iterative version of FGSM, it generates adversarial samples by adding small perturbations in the gradient direction multiple iterations and limiting the results to a certain range after each iteration [19], which can be formulated as

$$\mathbf{x}_{adv}^{(k)} = \Pi_\epsilon \left\{ \mathbf{x}_{adv}^{(k-1)} + \alpha \times \text{sign}\left(\nabla_{\mathbf{x}} L\left(f\left(\mathbf{x}_{adv}^{(k-1)}, y\right)\right)\right)\right\}, \quad (4)$$

where $k$ denotes the number of the iteration step and $\alpha$ is the step size of each iteration. $\Pi_\epsilon$ is used to ensure that the perturbation does not exceed a predefined range $\epsilon$.

For FGSM and PGD, we explore two scenarios: white-box and black-box attacks. In the white-box scenario, the attacker possesses full access to the model's topology, parameters, and gradients. Conversely, in the black-box scenario, the attacker is limited to basic information about the model. Without specific guidelines, we fix $\epsilon$ at 8/255 across all methods for testing. For iterative techniques PGD, the attack step is set at $\alpha = 0.01$, with a total of 7 steps.

## 2.3 Randomized Smoothing

As a strategy aimed at enhancing model adversarial robustness, randomized smoothing fortifies the model's defense against attacks through the inclusion of random noise [5]. In addition, there are many works that further explore randomized smoothing [16, 34, 40].

In a classification scenario mapping from $\mathbb{R}^d \rightarrow \mathcal{Y}$, randomized smoothing constitutes a method to formulate a refined classifier $g$ from any base classifier $f$. When evaluated at $x$, the smoothed classifier $g$ identifies the class that the base classifier $f$ is most inclined to predict when $x$ undergoes perturbation by isotropic Gaussian noise:

$$g(x) = \underset{c \in \mathcal{Y}}{\arg\max} \, \mathbb{P}\left(f(x + \epsilon) = c\right), \quad (5)$$

where $\epsilon \sim \mathcal{N}\left(0, \sigma^2 I\right)$. The noise level $\sigma$ serves as a hyperparameter for the smoothed classifier $g$, dictating a trade-off between adversarial robustness and accuracy.

## 3 Method

As aforementioned, we suggest achieving a better trade-off between adversarial robustness and accuracy for designing more practical SNN models. In this section, we first provide an empirical metric to quantify the trade-off between adversarial robustness and accuracy. We then introduce a new coding method called RSC to improve the adversarial robustness, which enhances the quantitative trade-off between adversarial robustness and accuracy. Furthermore, theoretical analysis is given to illustrate the conceptual equivalence of RSC and Poisson coding. After observing the inherent limitations of RSC, we further propose a specified training method designed for RSC to improve its clean accuracy and adversarial robustness.

## 3.1 Quantification Trade-off Estimation

Quantification trade-off is important for designing methods to better trade-off. Therefore, before giving full details of our methods, we first try to formulate a Quantification Trade-off Estimation.

**Definition 3.1.** *Quantification Trade-off Estimation (QTE).* From Figure 1, we can get that the process of achieving better trade-offs is also essentially making the area larger. For a trained model, assume

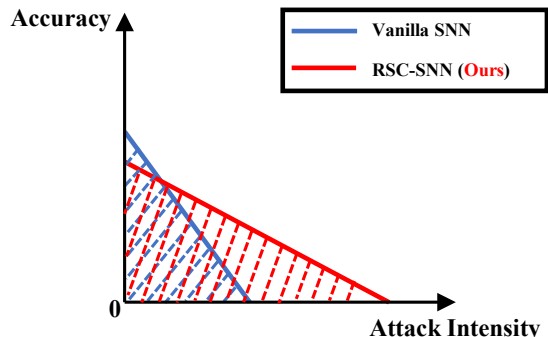

**Figure 1: An illustration of the trade-off between adversarial robustness and accuracy can be represented by the absolute value of the slope, indicating the SNN model's adversarial robustness. The area of a triangle can quantitatively estimate this trade-off. It is important to note that the slope illustrates the correlation between accuracy and attack strength, rather than implying a specific linear relationship.**

that its accuracy under attack intensity $\eta$ is $A(\eta)$, Quantification Trade-off Estimation between two attack intensities $\eta_a$ and $\eta_b$ can be formulated as

$$\text{QTE} = \left| \frac{(\eta_b - \eta_a)(A(\eta_b) + A(\eta_a))}{2} \right|. \quad (6)$$

Obviously, a larger Quantification Trade-off Estimation implies higher overall accuracy within the attack interval, signifying an improved trade-off in the model implementation. Simultaneously, it's noticeable that a smaller difference in $|\eta_b - \eta_a|$ corresponds to more accurate estimations of the model's quantification trade-offs.

## 3.2 Randomized Smoothing Coding (RSC)

Motivated by randomized smoothing, we first introduced random smoothing of Gaussian noise into the input of SNN. For each sample input $x \in \mathbb{R}^{3 \times H \times W}$, there is a given time step $T$. There are actually two ways to add noise to the input sample of the SNN model. One is to directly add a fixed noise $a$ to the input and no longer changes at each time step, which can be formulated as

$$\widetilde{x} = x + \epsilon, \epsilon \sim \mathcal{N}\left(0, \sigma^2 I\right), \quad (7)$$

it can also be expressed as

$$\widetilde{x} \sim \mathcal{N}\left(x, \sigma^2 I\right). \quad (8)$$

Another way is to add different Gaussian noises at each time step, which can be formulated as

$$\widetilde{x_i} = x_i + \epsilon_i, \epsilon_i \sim \mathcal{N}\left(0, \sigma^2 I\right), i = 1, \cdots, |T|, \quad (9)$$

where the noise level $\sigma$ serves as a hyperparameter for the model, dictating a trade-off between adversarial robustness and accuracy.

After adding noise to $x$, we further limit the range of $\widetilde{x}$ to $[0, 1]$, which can be formulated as

$$\widetilde{x}_{\text{clamp}} = \text{clamp}(\widetilde{x}) = \text{clamp}(x + \epsilon), \widetilde{x}_{\text{clamp}} \in [0, 1]. \quad (10)$$

For the first method, we named it RSC-I, and the second method RSC-II. In the experiment, we found that RSC-I can bring about a substantial improvement in adversarial robustness compared to RSC-II, and the degree of clean accuracy decrease will be higher than that of RSC-II; RSC-II can also bring about a small improvement in adversarial robustness, and the degree of clean accuracy decrease will be lower than that of RSC-I. Due to the need for higher adversarial robustness, we choose RSC-I as our main method.

## 3.3 Theoretical Analysis for Randomized Smoothing Coding

For the convenience of analysis, we first explain the meaning of the symbols. The input image is denoted as $x$. The expectation of $X$ is denoted as $E[X]$ and the covariance matrix is denoted as $\Sigma_X$.

For Poisson coding, the input is a random vector $X_P$, where the vector follows a Bernoulli binomial distribution with probability vector $p$. For randomized smoothing, the input is a random vector $X_{RS}$, which follows a normal distribution centered at $x$ with covariance $\sigma^2$, denoted as $X_{RS} \sim \mathcal{N}(x, \sigma^2)$. The following formula can be satisfied:

$$E[X_P] = E[X_{RS}] = x$$
$$\Sigma_{X_P} = \text{diag}(x(1-x)) = \text{diag}(x_1(1-x_1), \ldots, x_d(1-x_d)). \quad (11)$$
$$\Sigma_{X_{RS}} = \text{diag}(\sigma^2) = \text{diag}(\sigma_1^2, \sigma_2^2, \ldots, \sigma_d^2).$$

We can observe that Poisson coding and randomized smoothing coding can be considered as an equivalence in theory.

To better explain why randomized smoothing coding is better than Poisson coding, the results of the two different perturbed coding methods in a linear layer are explored through the following theorem. For the convenience of expression, we express the adversarial sample as $x + \epsilon$ and denote the linear layer as a deterministic weight matrix $W$, the original output as a random variable $Y_{P/RS_{original}}$ and the attacked output as a random variable $Y_{P/RS_{attack}}$. For both coding methods, the expectation of original $Y_{original}$ and attacked $Y_{attack}$ are given by:

$$E[Y_{original}] = E[WX] = WE[X] = Wx.$$
$$E[Y_{attack}] = W(x + \epsilon). \quad (12)$$

Although the expectations of both exhibit the same characteristics, their covariances behave differently. In Theorems 3.2 and 3.3, we obtain two covariance results with Poisson coding and randomized smoothing coding.

**Theorem 3.2.** *The covariance matrix of Poisson coding before and after the attack satisfies:*

$$\Sigma_{Y_{P_{original}}} = W\,diag(x(1-x))W^T.$$
$$\Sigma_{Y_{P_{attack}}} = W\,diag(x(1-x) + \epsilon(1-2x) - \epsilon^2)W^T. \quad (13)$$

**Theorem 3.3.** *The covariance matrix of randomized smoothing coding before and after the attack satisfies:*

$$\Sigma_{Y_{RS_{original/attack}}} = W\,diag(\sigma^2)W^T. \quad (14)$$

All the above theorems are proved in the supplementary material.

**Proposition 3.4.** *(Covariance invariance of RSC.) Poisson coding and randomized smoothing coding demonstrate distinct characteristics regarding covariance. For randomized smoothing coding, the*

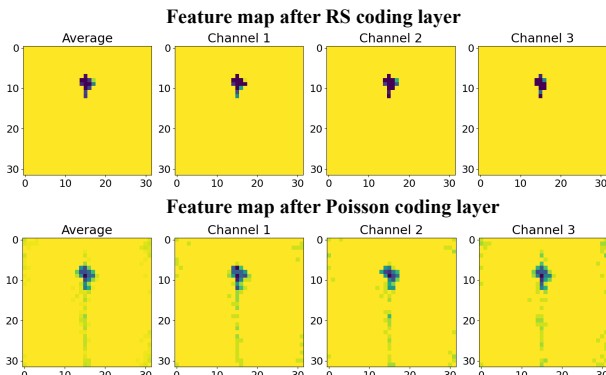

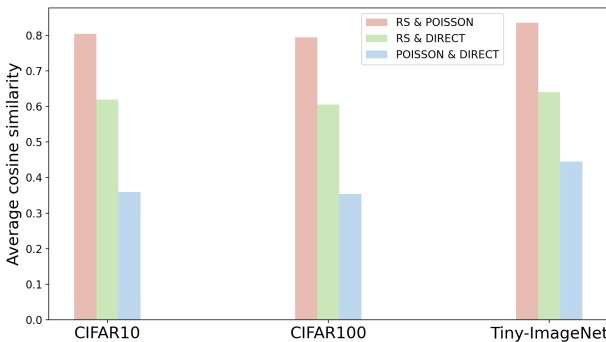

(a) An example of feature maps processed by RS and Poisson coding selected from CIFAR10, which shows a high degree of similarity between them.

(b) The average cosine similarity among three coding methods on CIFAR10, CIFAR100, and Tiny-ImageNet datasets indicates a notably high resemblance between randomized smoothing coding and Poisson coding.

**Figure 2: Visual verification of equivalence of randomized smoothing coding and Poisson coding.**

*noise variance introduced to the input remains constant and does not vary with the input itself.*

While Poisson coding and randomized smoothing share the high-level objective of enhancing adversarial robustness through noise averaging, they differ significantly in the specific nature of the noise added. Randomized smoothing employs isotropic Gaussian noise, resulting in predictable and manageable outcomes due to its uniform variance in all directions. In contrast, Poisson coding introduces noise whose covariance is influenced by both the input magnitude and perturbation attacks, leading to input-dependent variance. Consequently, the constant noise variance in randomized smoothing ensures a more stable and consistent smoothing process, potentially making it more effective in certain scenarios.

**Observation 3.5.** *(Equivalence between RSC and Poisson coding.) The image inputs to SNNs, following processing by the randomized smoothing coding layer, exhibit similar characteristics to that of the Poisson coding layer, displaying a notably high cosine similarity.*

The phenomenon in Figure 2(a) stems from coding properties. Randomized Smoothing regularizes feature maps by smoothing out irregularities, enhancing robustness. On the contrary, Poisson

coding, introduces binary spikes based on pixel intensity, potentially resulting in a noisier feature map. This observation is clearly depicted in Figure 2(a) and 2(b), where the average cosine similarity of randomized smoothing coding and Poisson coding stands at a notably elevated level, suggesting a equivalence between them. More visualization results are shown in the supplementary material.

**Table 1: Checklist for characteristic behaviors caused by obfuscated and masked gradients.**

| Items to identify gradient obfuscation | Test |
|---|---|
| (1) One-step attacks perform better than iterative attacks | Pass |
| (2) Black-box attacks are better than white-box attacks. | Pass |
| (3) Unbounded attacks do not reach 100% success. | Pass |
| (4) Increasing distortion bound does not increase success. | Pass |
| (5) Random sampling finds adversarial examples. | Pass |

**Table 2: Results show that our proposed method can still defend against EOT attacks.**

| Dataset | Architecture | Methods | Clean | PGD | EOTPGD |
|---|---|---|---|---|---|
| CIFAR10 | VGG-5 | RSC-0.1 | 80.29 | 37.28 | 28.72 |
| | VGG-5 | RSC-0.5 | 78.67 | 58.94 | 48.93 |
| CIFAR100 | VGG-11 | RSC-0.1 | 57.05 | 24.35 | 18.95 |
| | VGG-11 | RSC-0.5 | 56.25 | 33.96 | 27.14 |

## 3.4 Checks for RSC Gradient Obfuscation

Gradient obfuscation is the main reason why many adversarial defense methods are mistakenly considered effective. By certain methods, the neural network cannot produce accurate gradients, resulting in the inability to produce effective attacks.

To evaluate the attack effectiveness of the RSC , we employ the systematic checklist presented in [1] to scrutinize the gradient obfuscation of this novel coding scheme. This assessment is primarily grounded in the data delineated in Table 3 and Table 4 within the main body of the text, with a brief summary provided in Table 1. The detailed analysis can be found in the supplementary material.

Expectation over Transformation (EOT) computes the gradient over the expected transformation to the input as a method to attack randomized models [2]. We use EOTPGD proposed by Zimmermann [42] to evaluate the effectiveness of our method. The results in Table 2 show that our method can defend against EOT attacks.

## 3.5 Efficient-RSC Training (E-RSCT)

We found that while RSC improved the SNN model's adversarial robustness, it also led to a certain degree of decline in clean accuracy. To alleviate this problem, we propose E-RSCT for training SNN models using RSC. Motivated by the idea of knowledge distillation, we found that the corresponding ANN have naturally high clean accuracy, so we decided to use a pre-trained ANN as a teacher model to transfer the learned knowledge to the RSC-SNN model.

E-RSCT consists of two parts of loss functions during the training process. First, in order to transfer the learned knowledge from the

---

**Algorithm 1** Training process of E-RSCT for one epoch.

**Input**: An SNN to be trained with RSC; a hyperparameter $\sigma$; training dataset; total training iteration: $I_{\text{train}}$.

**Output**: The well-trained SNN.

1: **for** all $i = 1, 2, \ldots, I_{\text{train}}$ iteration **do**
2:     Get mini-batch training data, $x_{\text{in}}(i)$ and class label, $y(i)$;
3:     Feed the $x_{\text{in}}(i)$ into the SNN ;
4:     Generate new sample $\widetilde{x}$ by Eq. 7;
5:     Process $\widetilde{x}$ to get $\widetilde{x}_{\text{clamp}}$ by Eq. 10;
6:     Calculate the SNN output, $o_{\text{out}}(i)$ by Eq. 1 ;
7:     Compute the loss function $\mathcal{L}_{E-RSC} = \lambda \mathcal{L}_{KD} + \mathcal{L}_{P-S}$ by Eq. 16;
8:     Backpropagation and update model parameters;
9: **end for**

---

teacher model [13], we define the loss function of the first part as $\mathcal{L}_{KD}$ and use KL divergence to measure the difference between the student network and the teacher network to get $\mathcal{L}_{KD}$.

$$\mathcal{L}_{KD} = KL\left(O_{stu}, O_{tea}\right), \tag{15}$$

where $O_{tea}$ is the output of the teacher ANN model and $O_{stu}$ is the output of the student SNN model.

For the second part of the loss function, we use the pre-synaptic loss $\mathcal{L}_{P-S}$ [8] in training.

By combining $\mathcal{L}_{KD}$ and $\mathcal{L}_{P-S}$, we get the new loss function used for the novel training as follows

$$\mathcal{L}_{E-RSC} = \lambda \mathcal{L}_{KD} + \mathcal{L}_{P-S}, \tag{16}$$

where $\lambda$ is used to achieve a trade-off between $\mathcal{L}_{KD}$ and $\mathcal{L}_{P-S}$. Without special explanation, we select $\lambda = 0.1$ in the experiment. The detail of E-RSCT is shown in Algo.1.

## 4 Experiments

### 4.1 Experimental Setup

We verify the effectiveness of the proposed RSC and E-RSCT on multiple datasets and compare with ANNs simultaneously. In the following experiments, SNNs using direct coding are represented by DIRECT, SNNs using RS coding are represented by RSC and SNNs using Poisson coding are represented by POISSON. For the trade-off measurement of the model under different attacks, we use F-QTE to represent the Quantification Trade-off Estimation under the FGSM attack, and P-QTE to represent the Quantification Trade-off Estimation under the PGD attack. Detailed implementation is referred to in the supplementary material. For adversarial robustness evaluation, specific hyperparameter settings are introduced in Section 2.2. In the experiments of E-RSCT, we used the hyperparameters set in Section 3.5.

### 4.2 Performance under attacks.

**Clean accuracy and adversarial robustness.** Clean accuracy refers to the accuracy on the clean test dataset. It was represented as CLEAN in the experiment. The evaluation of adversarial robustness accuracy is denoted as FGSM and PGD respectively.

**Results on different datasets.** Table 3 illustrates the performance evaluation of our proposed RSC scheme. For constructing

**Table 3: White-box attack results on four datasets of ANN and three different coding methods of SNN. The best result is highlighted with bold and the second with underlined. The larger the better for all metrics.**

| Dataset | Architecture | Coding Methods | Clean | FGSM | PGD | F-QTE | P-QTE |
|---|---|---|---|---|---|---|---|
| | VGG-5 | ANN | 90.95 | 10.89 | 0.12 | 4.07 | 3.64 |
| | VGG-5 | Direct | 90.69 | 6.19 | 0.03 | 3.88 | 3.63 |
| CIFAR10 | VGG-5 | Poisson | 83.18 | 31.20 | 22.16 | 4.58 | 4.21 |
| | VGG-5 | RSC-0.1(**Ours**) | 80.29 | 51.29 | 37.28 | 5.26 | 4.70 |
| | VGG-5 | RSC-0.5(**Ours**) | 78.67 | **66.61** | **58.94** | **5.81** | **5.50** |
| | VGG-11 | ANN | 72.86 | 4.56 | 0.13 | 3.10 | 2.92 |
| | VGG-11 | Direct | 72.45 | 4.67 | 0.22 | 3.08 | 2.91 |
| CIFAR100 | VGG-11 | Poisson | 58.49 | 19.46 | 15.56 | 3.12 | 2.96 |
| | VGG-11 | RSC-0.1(**Ours**) | 57.05 | 32.48 | 24.35 | 3.58 | 3.22 |
| | VGG-11 | RSC-0.2(**Ours**) | 55.42 | 37.83 | 29.69 | **3.73** | 3.40 |
| | VGG-11 | RSC-0.5(**Ours**) | 51.30 | **40.81** | **33.96** | 3.68 | **3.41** |
| | VGG-16 | ANN | 60.77 | 2.08 | 0.00 | 2.51 | 2.43 |
| | VGG-16 | Direct | 57.90 | 2.04 | 0.01 | 2.40 | 2.32 |
| Tiny-ImageNet | VGG-16 | Poisson | 48.14 | 6.79 | 2.68 | 2.20 | 2.03 |
| | VGG-16 | RSC-0.01(**Ours**) | 48.33 | 7.73 | 2.15 | 2.24 | 2.02 |
| | VGG-16 | RSC-0.1(**Ours**) | 47.47 | **22.63** | **13.75** | **2.80** | **2.45** |
| | ResNet-19 | ANN | 67.00 | 0.66 | 0.00 | 2.71 | **2.68** |
| | ResNet-19 | Direct | 56.41 | 2.57 | 0.02 | 2.36 | 2.26 |
| | ResNet-19 | Poisson | 40.21 | 10.61 | 2.68 | 2.03 | 1.72 |
| ImageNet | ResNet-19 | RSC-0.1(**Ours**) | 44.25 | 17.73 | **8.50** | 2.48 | 2.11 |
| | SEW-ResNet-18 | Direct | 64.40 | 4.56 | 0.00 | 2.76 | 2.58 |
| | SEW-ResNet-18 | Poisson | 52.29 | 15.73 | 4.70 | 2.72 | 2.28 |
| | SEW-ResNet-18 | RSC-0.1(**Ours**) | 53.79 | **25.86** | 7.38 | **3.19** | 2.45 |

**Table 4: Black-box attack results on four datasets of ANN and three different coding methods of SNN. The best result is highlighted with bold and the second with underlined. The larger the better for all metrics.**

| Dataset | Architecture | Coding Methods | Clean | FGSM | PGD | F-QTE | P-QTE |
|---|---|---|---|---|---|---|---|
| | VGG-5 | Direct | 90.69 | 20.75 | 3.52 | 4.46 | 3.77 |
| CIFAR10 | VGG-5 | Poisson | 83.18 | 43.06 | 36.15 | 5.05 | 4.77 |
| | VGG-5 | RSC-0.1(**Ours**) | 80.29 | **59.25** | **49.55** | **5.58** | **5.19** |
| | VGG-11 | Direct | 72.45 | 11.79 | 4.08 | 3.37 | 3.06 |
| CIFAR100 | VGG-11 | Poisson | 58.49 | 31.03 | 27.36 | 3.58 | 3.43 |
| | VGG-11 | RSC-0.1(**Ours**) | 57.05 | **41.74** | **35.56** | **3.95** | **3.70** |
| | VGG-16 | Direct | 57.90 | 14.82 | 8.15 | 2.91 | 2.64 |
| Tiny-ImageNet | VGG-16 | Poisson | 48.14 | 21.22 | 16.73 | 2.77 | 2.59 |
| | VGG-16 | RSC-0.1(**Ours**) | 47.47 | **35.06** | **29.40** | **3.30** | **3.07** |
| | SEW-ResNet-18 | Direct | 64.40 | 15.75 | 11.89 | 3.21 | 3.05 |
| ImageNet | SEW-ResNet-18 | Poisson | 52.29 | 16.87 | 16.65 | 2.77 | 2.76 |
| | SEW-ResNet-18 | RSC-0.1(**Ours**) | 53.79 | **29.52** | **27.91** | **3.33** | **3.27** |

effective attacks on SNN, all gradient attacks are applied based on BPTT. The specific implementation of BPTT is in the supplementary material. The results consistently demonstrate the efficacy of our RSC in enhancing model adversarial robustness, as evidenced by notable enhancements in adversarial robustness accuracy across all attack methodologies. Particularly striking is the significant improvement in adversarial robustness against stronger white-box iterative attacks. Notably, the VGG-5 model exhibited a remarkable 58.91% increase in accuracy when attacked by PGD compared to direct coding on the CIFAR-10 dataset. Simultaneously, the improvement of F-QTE and P-QTE also shows that our method better achieves the trade-off between accuracy and adversarial robustness. The experimental results that RSC shows more adversarial robustness than Poisson coding well verify Theorem 3.2 and 3.3.

**Experiments on Tiny-ImageNet and ImageNet.** Previous work was limited to small datasets such as CIFAR-10 and CIFAR-100. We also conducted experiments on Tiny-ImageNet and ImageNet to provide a baseline for the evaluation of SNN adversarial robustness.

**Table 5: Enhanced robustness with adversarial training. The best result is highlighted with bold. The larger the better for all metrics.**

| Dataset | Architecture | Methods | Clean | FGSM | PGD |
|---|---|---|---|---|---|
| CIFAR10 | VGG-5 | RSC-0.1 | 80.29 | 51.29 | 37.28 |
| | VGG-5 | RSC-0.1 + Adv | 80.27 | **55.58** | **43.59** |
| CIFAR100 | VGG-11 | RSC-0.1 | 57.05 | 32.48 | 24.35 |
| | VGG-11 | RSC-0.1 + Adv | 56.25 | **35.53** | **27.57** |

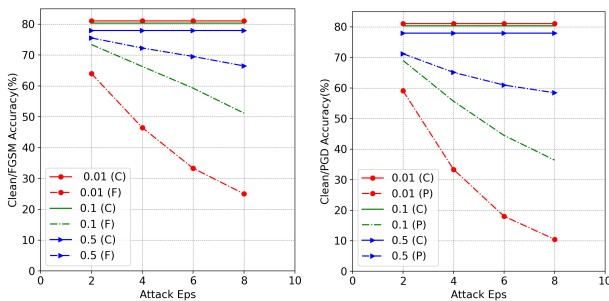

(a) Ablation experiments of noise $\sigma^2$ - FGSM.

(b) Ablation experiments of noise $\sigma^2$ - PGD.

**Figure 3: Ablation experiment for noise level $\sigma^2$.**

It can be seen that the adversarial robustness of RSC-SNN has been significantly improved compared to DIRECT and POISSON.

**Black-box attack results on different datasets.** In this section, we assess the adversarial robustness of RSC against black-box attacks. We utilize a separately trained SNN with an identical architecture to generate white-box attack samples. The results presented in Table 4 demonstrate that RSC demonstrates significant resilience against adversarial attacks, surpassing traditional SNNs in both FGSM and PGD scenarios. This robustness positions RSC as notably more resilient to adversarial intrusions in black-box scenarios. Compared with traditional SNNs, RSC-SNN's F-QTE and P-QTE have also improved in black-box attack scenarios.

**Enhanced Robustness with Adversarial Training.** Adversarial training, has been the most widely accepted defense method. To evaluate the combined robustness of RSC and adversarial training, we explored their transferability and scalability. We trained SNN-Direct, SNN-Poisson and RSC using low-intensity FGSM samples and then exposed them to more complex and larger $\epsilon$ white-box attacks. Results in Table 5 (rows 1-2 for CIFAR10 and rows 3-4 for CIFAR100) show a significant boost in robustness for RSC when combined with adversarial training. On CIFAR-10, this combination increased robustness for FGSM (to 55.58% from 51.29%) and PGD (to 43.59% from 37.28%). For CIFAR-100, resilience improved against FGSM (to 35.53% from 32.48%) and PGD (to 27.57% from 24.35%). In conclusion, the integration of RSC with adversarial training gives it the versatility to withstand a wider range of more powerful adversarial attacks.

### 4.3    The effectiveness of E-RSCT.

We can see from Table 3 that while RSC brings significant adversarial robustness improvement to the model, it also leads to clean accuracy decline. To alleviate this problem, we proposed E-RSCT for RSC-SNN. Table 6 summarizes the clean and robust accuracy of SNN models trained with and without E-RSCT. It can be seen from Table 6 that the clean accuracy and adversarial robustness of the model trained using E-RSCT on all datasets have been improved to a certain extent. On the CIFAR10 dataset, while the clean accuracy was improved by 1.74%, its robust accuracy for FGSM and PGD was also improved by 3.23% and 2.70% respectively. On other datasets, an average accuracy improvement of more than 1% has been achieved. The experimental results concretely verify the effectiveness of our proposed training algorithm. The improvement of F-QTE and P-QTE also shows that E-RSCT achieves better trade-offs.

### 4.4    Ablation Studies

**Effect of different noise levels $\sigma^2$ on RSC.** Investigating the pivotal role of the noise level parameter $\sigma^2$ within the novel introduced RSC framework holds significant importance in regulating the model's adversarial robustness. We conducted an exhaustive ablation study to discern the effect of parameter variations on the model's adversarial robustness. Specifically, we selected three distinct values: $\sigma^2 = 0.01$, $\sigma^2 = 0.1$, and $\sigma^2 = 0.5$ to meticulously evaluate the model's adversarial robustness under varying attack intensities FGSM and PGD across multiple values ($\epsilon = 2, 4, 6, 8$). The comprehensive experimental outcomes are visually presented in the Figure 3. The solid line shows clean accuracy, and the dotted line shows post-attack accuracy.

From Figure 3, a noticeable trend emerges: as the noise level ($\sigma^2$) escalates, there's a concurrent decline in the model's clean accuracy, a phenomenon congruent with our empirical analyses. Simultaneously, with an increase in $\sigma^2$, the model's adversarial robustness exhibits a consistent uptrend. Remarkably, the robust accuracy under various attack intensities showcases a discernible hierarchy, wherein higher $\sigma^2$ values correspond to augmented adversarial robustness. Specifically, the variations in robust accuracy across different attack intensities become more pronounced with increasing $\sigma^2$. Evidently, this underscores the significance of striking a balance between clean accuracy and adversarial robustness in real-world RSC applications. Achieving this balance necessitates meticulous exploration of $\sigma^2$ values to pinpoint the optimal choice aligned with specific application requisites.

### 4.5    Comparison with State-of-the-art Work on Adversarial Robustness of SNN

To evaluate the effectiveness of our proposed RSC, we compare it with the results of existing state-of-the-art work. We conduct experimental comparisons in the case of FGSM and PGD white-box attacks on the CIFAR-10 and CIFAR-100 datasets. The hyperparameters of the attack are set to $\epsilon = 8/255$ for FGSM and $\alpha = 0.01$ for PGD with a total of 7 steps. The comparison results are shown in Table 7.

**CIFAR-10.** On the CIFAR-10 dataset, employing the RSC VGG-5 model with a noise level of $\sigma^2 = 0.1$, our approach showcased

**Table 6: Comparison of results with E-RSCT and without E-RSCT. The best result is highlighted with bold. The larger the better for all metrics.**

| Dataset | Architecture | Methods | Clean | FGSM | PGD | F-QTE | P-QTE |
|---------|-------------|---------|-------|------|-----|-------|-------|
| CIFAR-10 | VGG-5 | Baseline | 80.29 | 51.29 | 37.28 | 5.26 | 4.70 |
| | VGG-5 | +E-RSCT | **82.03** | **54.52** | **39.98** | **5.46** | **4.88** |
| CIFAR-100 | VGG-11 | Baseline | 57.05 | 32.48 | 24.35 | 3.58 | 3.26 |
| | VGG-11 | +E-RSCT | **58.04** | **34.89** | **26.67** | **3.72** | **3.39** |
| Tiny-ImageNet | VGG-16 | Baseline | 47.47 | 22.63 | 13.75 | 2.80 | 2.45 |
| | VGG-16 | +E-RSCT | **48.29** | **24.01** | **15.46** | **2.89** | **2.55** |
| ImageNet | SEW-ResNet-18 | Baseline | 53.79 | 25.86 | 7.38 | 3.19 | 2.45 |
| | SEW-ResNet-18 | +E-RSCT | **54.77** | **27.21** | **8.84** | **3.28** | **2.54** |

**Table 7: Comparison with others works. The best result is highlighted with bold and the second with underlined. The larger the better for all metrics.**

| Dataset | Architecture | Methods | FGSM | PGD | F-QTE | P-QTE | Clean |
|---------|-------------|---------|------|-----|-------|-------|-------|
| CIFAR10 | VGG-5 | Baseline | 6.19 | 0.03 | 3.88 | 3.63 | 90.69 |
| | VGG-5 | Sharmin et al. [28][ECCV] | 15.00 | 3.80 | 4.17 | 3.72 | 89.30 |
| | VGG-5 | Kundu et al. [15][ICCV] | 38.00 | 9.10 | 5.02 | 3.86 | 87.50 |
| | VGG-5 | Ding et al. [9][NeurIPS] | 45.23 | 21.16 | 5.44 | 4.48 | 90.74 |
| | VGG-5 | **Our work** | **54.52** | **39.98** | **5.46** | **4.88** | 82.03 |
| CIFAR100 | VGG-11 | Baseline | 5.30 | 0.02 | 3.15 | 2.93 | 73.33 |
| | VGG-11 | Sharmin et al. [28][ECCV] | 15.50 | 6.30 | 3.20 | 2.83 | 64.40 |
| | VGG-11 | Kundu et al. [15][ICCV] | 22.00 | 7.50 | 3.48 | 2.90 | 65.10 |
| | VGG-11 | Ding et al. [9][NeurIPS] | 25.86 | 10.38 | **3.87** | 3.25 | 70.89 |
| | VGG-11 | **Our work** | **34.89** | **26.67** | 3.72 | **3.39** | 58.04 |
| Tiny-ImageNet | VGG-16 | Baseline | 2.04 | 0.03 | 2.40 | 2.32 | 57.90 |
| | VGG-16 | **Our work** | **24.01** | **15.46** | **2.89** | **2.55** | 48.29 |
| ImageNet | SEW-RESNET-18 | Baseline | 4.56 | 0.00 | 2.76 | **2.58** | 64.40 |
| | SEW-RESNET-18 | **Our work** | **27.21** | **8.84** | **3.28** | 2.54 | 54.77 |

promising advancements. As indicated in the table, our method notably elevated the model accuracy against FGSM and PGD attacks by 48.33% and 39.95%, respectively, in contrast to the vanilla model. Furthermore, in comparison with the best-performing outcomes [9], our approach achieved a substantial enhancement in accuracy by 9.29% for FGSM and 18.82% for PGD attacks.

**CIFAR-100.** On the CIFAR-100, our utilization of the RSC VGG-11 model with a noise level of $\sigma^2 = 0.1$ yielded notable advancements. The provided table demonstrates that our method elevated accuracy against FGSM and PGD attacks by 29.59% and 26.65%, respectively, surpassing the performance of the vanilla model. In comparison to the top-performing outcomes, our approach showcased compelling enhancements, achieving a remarkable 9.03% accuracy improvement for FGSM and 16.29% for PGD attacks.

Upon comparing our experimental outcomes with the State-of-the-Art (SOTA) approaches on the CIFAR-10 and CIFAR-100 datasets, our proposed method showcased significant enhancements in the model's adversarial robustness. F-QTE and P-QTE also achieve comparable results to SOTA. Notably, while augmenting adversarial robustness, we observed a decline in clean accuracy compared to the vanilla model. Hence, to achieve a balance between clean accuracy and adversarial robustness, we meticulously fine-tuned various hyperparameters involved in the training process.

Parameters like the noise level $\sigma$, the proportion of the loss function $\lambda$, etc., were judiciously adjusted in the E-RSCT framework for refinement. However, achieving superior results still necessitates further extensive research and exploration.

## 5 Conclusion

Our present work offers a theoretical foundation for the observed empirical robustness of Poisson-encoded classifiers against adversarial attacks. Observing that randomized smoothing and Poisson coding exhibit similar characteristics, we first demonstrate the equivalence between the two, explaining why Poisson coding SNNs have adversarial robustness. Furthermore, our theoretical analysis shows that randomized smoothing is more stable than Poisson coding. On this basis we propose a novel randomized smoothing coding, which enhances the adversarial robustness of SNNs. Experimental results show that our method achieves state-of-the-art adversarial robustness. However, there is still room for further improvement in clean accuracy. Therefore, valuable future work directions include further improving the trade-off between adversarial robustness and clean accuracy. We believe our work will pave the way for further research on more applications of safety-critical SNNs.

## Acknowledgments

This work was partially supported by the National Distinguished Young Scholars (62325603), the National Natural Science Foundation of China (62236009,U22A20103,62441606), the Beijing Natural Science Foundation for Distinguished Young Scholars (JQ21015), the China Postdoctoral Science Foundation (GZB20240824, 2024M753497), and the CAAI-MindSpore Open Fund, which was developed on the OpenI Community.

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
