# OpenReview forum: "RSC-SNN: Exploring the Trade-off Between Adversarial Robustness and Accuracy in Spiking Neural Networks via Randomized Smoothing Coding"
_acmmm.org/ACMMM/2024/Conference — MM2024 Poster_

### Official Review · Reviewer_KWo8 · 2024-04-29

**Rating:** 2
**Confidence:** 3

**Summary:**

This paper claims to offer a theoretical analysis of the equivalence between Poisson coding and randomized smoothing. Additionally, it employs randomized smoothing to train SNNs and observes improved robustness. Inspired by knowledge distillation, the paper suggests using an ANN as a teacher to train SNNs and devises a corresponding loss function.

**Strengths:**

(1) Randomized smoothing is a well-known technique for enhancing the robustness of ANNs, but it has not yet been applied to SNNs. This paper makes the first attempt to transfer this technique to SNNs.

(2) Training SNNs using knowledge distillation has not been explored yet.

**Limitations:**

Q1: This paper appears to overstate its theoretical contributions. The authors merely calculate the expectation and variance without providing rigorous mathematical proof of the equivalence between Poisson coding and random smoothing.

Q2: Why is there a linear relationship between classification accuracy and attack intensity, as shown in Fig.2? Is there any therotical or experimental proof of this relationship? I disagree with this assertion. Consequently, I believe the proposed QTE is not justified.

Q3: Thr.3.2 and 3.3 are based solely on a simple linear layer and do not include an analysis of fundamental firing functions or other neuronal dynamics. This makes the paper less relevant to SNN.

Q4: The paper lacks details on critical attack parameters. Specifically, what are the $\ell_\infty$ bounds for FGSM and PGD? Additionally, how many iterations are used in these attacks?

Q5: In Fig.4, the distinction between clean accuracy and FGSM/PGD accuracy is unclear from the caption. Based on the context, I assume that the dotted lines represent clean accuracy. Given this assumption, it's unusual that the FGSM/PGD accuracy is significantly higher than the clean accuracy. This requires further clarification.

Q6: This paper presents contradictory claims: on one hand, it asserts that Poisson coding is equivalent to randomized smoothing; on the other hand, it suggests that randomized smoothing is more stable than Poisson coding. These conclusions seem to conflict. If the two strategies are truly equivalent, then Poisson coding could be represented as a form of randomized smoothing with a specific $\delta$.

**Suitability:**

2

---

### Official Review · Reviewer_kQqB · 2024-05-11

**Rating:** 3
**Confidence:** 4

**Summary:**

The author aims to enhance the adversarial robustness of spiking neural networks (SNNs) by studying the spiking encoding method. They prove the equivalence of Poisson coding and stochastic smoothing as the source of SNN's robustness. They propose Randomized Smoothing Coding and Efficient-RSC Training to improve SNN's robustness and clean accuracy. Experiments show that RSC-SNN enhances adversarial robustness compared to traditional ANN and SNN. The paper concludes that SNNs can maintain their robustness advantage on large-scale tasks.

**Strengths:**

The paper explores the trade-off between adversarial robustness and accuracy in Spiking Neural Networks through Randomized Smoothing Coding.

The topics are relevant to ACMMM.

The performance improvement is clear on various benchmarks.

**Limitations:**

The choice of noise levels (0.01, 0.1, 0.5) is here questioned as no information on the choice of the value.

The main contributions seem to be applications of existing techniques. Comparisons with previous works that defended against adversarial noise through input transformations are deemed insufficient.

The paper lacks check for gradient obfuscation, which could mistakenly attribute effectiveness to the defense method.

The advantage of Randomized Smoothing Coding over Poisson coding is unclear, especially since Poisson coding is more hardware-friendly for SNNs.

Using RSC may lead to performance degradation, which should be discussed in the conclusion.

**Suitability:**

2

---

### Official Review · Reviewer_FQtt · 2024-05-22

**Rating:** 4
**Confidence:** 2

**Summary:**

A method of Randomized Smoothing Coding was proposed, and it was demonstrated that Poisson coding and randomized smoothing coding can be theoretically considered equivalent.

**Strengths:**

1. Figure 1 clearly illustrates the distinctions between different encodings.
2. Additionally, the perspective that Poisson coding and randomized smoothing coding can be theoretically considered equivalent is further explored. The reasons why randomized smoothing coding may be superior to Poisson coding are explained on a theoretical level, with great detail and logical coherence.

**Limitations:**

1. In line 151, the term "trade- off" has an extra space in the middle.
2. Regarding Figure 3(a), a more comprehensive explanation is required for the feature maps after different encoding layers. For instance, why does the feature map after the RS encoding layer appear cleaner compared to that after Poisson encoding?
3. The explanation of "white-box and black-box attacks" in the text remains unclear to the reviewer.

**Suitability:**

3

---

### Official Review · Reviewer_ojjC · 2024-05-24

**Rating:** 4
**Confidence:** 3

**Summary:**

This paper proposed a method named Randomized Smoothing Coding (RSC) to enhance the robustness and performance of SNNs. The authors also provided theorems and empirical observation to discuss the equivalence between RSC and traditional Possion coding from a mathematical perspective. Experimental results under different attack scenarios validated the effectiveness of RSC.

**Strengths:**

1. The theoretical analysis of this paper is solid.
2. The authors have made numerous experiments to validate the effectiveness of their proposed method.

**Limitations:**

1. The theme of this paper is not very suitable for the MM community.
2. Considering that the robustness of SNNs is a relatively niche topic in the SNN community, the contribution of this work may not be very significant.

**Suitability:**

2

---

### Meta-Review · Area_Chair_dHcT · 2024-07-01

**Recommendation:** Accept (Poster)
**Confidence:** 4

**Metareview:**

This paper introduces Randomized Smoothing Coding (RSC) to improve the adversarial robustness and performance of spiking neural networks (SNNs). All the reviewers agreed that this paper presents an improvement in the robustness and performance of SNNs. Three reviewers highlighted the solid theoretical contributions of the paper, particularly the analysis demonstrating the equivalence between Poisson coding and randomized smoothing coding. Two reviewers also found that the proposed methods show promising quantitative results.

One reviewer noted the lack of comparison with related work. Other weaknesses included the niche focus on SNN robustness and insufficient explanations of critical attack parameters and terms like "white-box and black-box attacks."

I tend to accept this paper. However, the authors should include EOT experiments in the final version.